# SPL12 Regulates *AGL6* and *AGL21* to Modulate Nodulation and Root Regeneration under Osmotic Stress and Nitrate Sufficiency Conditions in *Medicago sativa*

**DOI:** 10.3390/plants11223071

**Published:** 2022-11-13

**Authors:** Vida Nasrollahi, Ze-Chun Yuan, Susanne E. Kohalmi, Abdelali Hannoufa

**Affiliations:** 1Agriculture and Agri-Food Canada, 1391 Sandford Street, London, ON N5V 4T3, Canada; 2Department of Biology, University of Western Ontario, 1151 Richmond Street, London, ON N6A 3K7, Canada

**Keywords:** *Medicago sativa*, *SPL12*, *AGL6*, *AGL21*, nodulation, osmotic stress, nitrate

## Abstract

The highly conserved plant microRNA, miR156, affects root architecture, nodulation, symbiotic nitrogen fixation, and stress response. In *Medicago sativa*, transcripts of eleven *SQUAMOSA-PROMOTER BINDING PROTEIN-LIKE*, *SPL*s, including *SPL12*, are targeted for cleavage by *miR156*. Our previous research revealed the role of SPL12 and its target gene, *AGL6*, in nodulation in alfalfa. Here, we investigated the involvement of SPL12, AGL6 and AGL21 in nodulation under osmotic stress and different nitrate availability conditions. Characterization of phenotypic and molecular parameters revealed that the SPL12/AGL6 module plays a negative role in maintaining nodulation under osmotic stress. While there was a decrease in the nodule numbers in WT plants under osmotic stress, the *SPL12*-RNAi and *AGL6*-RNAi genotypes maintained nodulation under osmotic stress. Moreover, the results showed that SPL12 regulates nodulation under a high concentration of nitrate by silencing *AGL21*. *AGL21* transcript levels were increased under nitrate treatment in WT plants, but *SPL12* was not affected throughout the treatment period. Given that *AGL21* was significantly upregulated in *SPL12*-RNAi plants, we conclude that SPL12 may be involved in regulating nitrate inhibition of nodulation in alfalfa by targeting *AGL21*. Taken together, our results suggest that SPL12, AGL6, and AGL21 form a genetic module that regulates nodulation in alfalfa under osmotic stress and in response to nitrate.

## 1. Introduction

Whereas root system architecture in plants is critical, because of its role in controlling nutrient cycling, water use efficiency, and resistance to biotic and abiotic stresses, balanced nutritional elements can also be helpful in combatting environmental stress [1,2]. The availability of nitrogen is a major limiting factor determining crop growth and productivity [3]. Although synthetic nitrogen fertilizers have increased crop yields, they are disadvantaged by their significant human health and environmental costs [4]. Among nitrogen-fixing crops, alfalfa (*Medicago sativa* L.) is the most widely cultivated forage crop around the world [5], and is one of the most important crops commercially grown for various purposes, including crop rotations, livestock feed, and soil improvement. This is due in part to its ability to form a symbiotic relationship with rhizobial bacteria [6,7,8]. The symbiotic nitrogen fixation of legumes takes place in root nodules [9], where rhizobia reduce atmospheric di-nitrogen to ammonia by expressing the nitrogenase enzyme [10]. 

As nitrogenase is exceptionally rich in sulfur [11], this element becomes limiting during symbiosis. There is a high demand for sulfur in nodulated legumes, and nitrogen fixation is more sensitive to sulfur deficiency than to nitrate uptake [12]. An abundant supply of sulfur in plants markedly increases nodulation and nitrogen fixation [12,13]. On the other hand, sulfur-deficiency in plants leads to a decrease in nodulation, nodule metabolism, and nitrogenase biosynthesis and activity, presumably because of the low-availability of sulfur-containing cysteine and methionine [14]. In *Lotus japonicus*, the *LjSST1* (*SYMBIOTIC SULFATE TRANSPORTER1*) gene encodes a sulfate transporter that is specifically and highly expressed in the nodules, suggesting a major role in the transport of sulfate from the plant to the bacteroids [15]. In *Medicago truncatula,* a Group 3 SULTR (*MtSULTR3.5*) is strongly expressed in the nodules [16], and its expression is upregulated in roots subjected to salt stress [17]. Interestingly, *SULTR3.1* and *SULTR3.4* are upregulated in the roots of both Arabidopsis and *M. truncatula* plants subjected to drought stress. 

Research has shown that microRNA156 (miR156) regulates the response to abiotic stress, including heat [18], salinity [19], and drought [20,21] in alfalfa. miR156 targets a number of *SPL* genes for post-transcriptional silencing in various plant species [22,23,24]. Of the known SPLs in alfalfa, SPL13, SPL9, and SPL8 have been investigated for their role in drought tolerance in alfalfa, where their downregulation led to improved tolerance [20,21,25,26]. Moreover, a previous study found that overexpression of miR156 enhanced nodule numbers and nitrogenase activity in alfalfa [27,28]. Most recently, we showed that miR156-targeted *SPL12* and its downstream target *AGL6* are involved in regulation of nodulation in this plant [29].

The MADS (MCM1/AGAMOUS/DEFICIENS/SRF) box proteins are a family of transcription factors that participate in many aspects of plant development and morphogenesis, including floral organ speciation, fruit development, and root development [30,31,32,33,34,35]. Recently, many studies have shown that MADS-box transcription factors play an important role in the regulation of plant responses to extreme conditions, like drought, salt, and heat and cold stresses [36,37,38]. In alfalfa, a total of 120 MADS-box genes have been identified [38]. Of the known MADS-box genes in alfalfa, *MsMADS001*, *MsMADS075*, and *MsMADS090* were significantly induced by drought and salt stresses [38], suggesting that these genes might regulate tolerance to multiple stresses in alfalfa. Moreover, MADS-box transcription factors are also involved in root architecture changes in response to alterations in nitrogen supply. For example, ANR1 (Arabidopsis NITRATE REGULATED1) was the first MADS-box transcription factor shown to stimulate lateral root development in response to nitrate availability [39,40]. AGL21, which is highly expressed in lateral root primordia [41], was found to control lateral root development by regulating genes associated with auxin biosynthesis in Arabidopsis [41]. In rice, *Os*MADS25, an ANR1-like protein, positively regulates lateral and primary root development by promoting nitrate accumulation, and increasing the expression of nitrate transporter genes at high nitrate concentrations [41]. Huang et al. [42] showed that OsMADS57, a rice MADS-box transcription factor, is a positive regulator of a high-affinity nitrate transporter gene (*OsNRT2.3a*). 

In the current study, we investigated the role of SPL12 in root architecture and nodulation under osmotic stress and high nitrate concentrations using RNAi-silenced *SPL12* (*SPL12*-RNAi) alfalfa plants. We also investigated the involvement of SPL12 target genes, *AGL6* and *AGL21* in nodule regulation under stress conditions in alfalfa.

## 2. Results

### 2.1. SPL12 Silencing Attenuates the Effect of Nitrate on Nodulation

Nitrogen abundance in the soil inhibits nodulation, and this regulatory process is a part of the autoregulation of the nodulation (AON) pathway [43,44]. Given the effects of SPL12 on nodulation, we assessed whether the nodulation capacity of *SPL12*-RNAi transgenic plants (RNAi12-7, RNAi12-24, and RNAi12-29) was affected by nitrate treatment. The number of nodules was compared between WT and *SPL12*-RNAi plants treated with 3 mM, 8 mM, and 20 mM of potassium nitrate (KNO_3_) or potassium chloride (KCl) at 21 days after inoculation (dai). There were no significant changes in nodulation between the plants that were watered with 3 mM KCl or KNO_3_ (Appendix A). All plants that were watered with KCl (8 mM and 20 mM) formed active nitrogen-fixing nodules that were pink-colored (containing leghaemoglobin) (Figure 1A, Appendix A), with no significant difference in the numbers of either white or pink nodules between *SPL12*-RNAi and WT plants (Appendix A). When watered with 8 mM KNO_3_, all of the *SPL12*-RNAi plants formed significantly more mature pink nodules relative to WT (Figure 1B). Under the 20 mM KNO_3_ treatment, WT plants formed only small white nodules, while RNAi12-24 and RNAi12-29 plants produced mainly pink nodules (Figure 1C). These results indicate that silencing of *SPL12* prevents nitrate-inhibition nodulation in alfalfa.

### 2.2. Effect of Nitrate on Expression of SPL12 and AGL21 

To shed light on the possible role of SPL12 in nitrate inhibition of nodulation, we determined whether the transcript levels of *SPL12* and *AGL21* were regulated by nitrate. *AGL21* in alfalfa is closely related to the ANR1 clade in Arabidopsis (Figure 2A), and *At*ANR1, a member of the ANR1 clade, is involved in the nitrate regulation of root development in Arabidopsis [39,40]. Additionally, *AtAGL21*, another member of this clade, is upregulated by nitrogen deprivation [41]. Based on these findings, we hypothesized that AGL21 is involved in the nitrate regulation of nodulation in alfalfa. To test this hypothesis, we investigated changes in the transcript levels of *AGL21* and *SPL12* under nitrate treatment (Figure 2B). The results showed that the transcript levels of *AGL21* were increased after 5 h of nitrate treatment in WT plants, but the transcript levels of *SPL12* was not affected throughout the treatment period. Based on these results, we propose that SPL12 is involved in regulating nitrate inhibition of nodulation in alfalfa by targeting *AGL21*.

To investigate the expression profile of *SPL12* in alfalfa, we measured its transcript levels in various organs (leaf, stem, and root) of 21-day-old WT alfalfa plants. The transcript levels of *SPL12* were detected at similar levels in all three tissues (Figure 3A). The transcript levels of *AGL21* were also determined in the same tissues, and found to be nearly undetectable in leaf and stem tissues and highly expressed in roots (Figure 3B). The low level of *AGL21* in the leaf is consistent with previous reports that the Arabidopsis *ANR1-like* genes were expressed primarily in roots [45]. *AGL79* expression was also nearly undetectable in leaf tissues in Arabidopsis [46].

### 2.3. SPL12 Is a Direct Regulator of AGL21

Previous transcriptomic analysis of miR156 overexpression (miR156-OE) plant A17, revealed 8,373 differentially expressed genes between roots of WT and miR156-OE [27]. Of the many genes differentially expressed in miR156-OE plant A17 relative to WT, *AGL21* (MS.gene069166, MS.gene068633, MS.gene70086, and MS.gene027842), a gene that encodes a yet-to-be characterized alfalfa MADS box protein was significantly upregulated in A17 [27]. This gene had also significantly high transcript levels in *SPL12*-RNAi based on the transcriptomic analysis of *SPL12*-RNAi genotypes [29], which was confirmed by RT-qPCR (Figure 4A). These results suggested that *AGL21* might be regulated by SPL12, thus, further characterization was carried out by ChIP-qPCR using *35S::SPL12m-GFP* plants to determine if *AGL21* is a direct target of SPL12. DNA sequence analysis revealed that the promoter (2000 bp) of alfalfa *AGL21* has four putative SPL binding sequences with a core GTAC sequence that are distributed in three regions (I, II, III) (Figure 4B), and three of them (in regions I and III) possess the typical NNGTACR SBP-binding consensus (Appendix A). The three regions were tested for SPL12 occupancy using ChIP-qPCR analysis of *35S::SPL12m-GFP* plants. Compared to WT, *35S::SPL12m-GFP* plants showed significantly higher SPL12 binding to the *AGL21* promoter region (Figure 4C), and occupancy in the three regions was substantially higher than that at the negative control *LOB1* (Figure 4C), indicating that the SPL12 protein can bind to multiple regions in the *AGL21* promoter to regulate its expression.

### 2.4. Effect of SPL12 Silencing in Response to Osmotic Stress

The involvement of miR156 in regulating drought and salinity responses was previously demonstrated in alfalfa [20,21]. To determine whether *SPL12* is regulated in response to osmotic stress, the *SPL12* transcript levels were assessed in five-week-old WT alfalfa plants treated with mannitol (400 mM) for two weeks. The transcript abundance of *SPL12* was significantly increased (1.4 fold) under osmotic stress compared to the well-watered control treatment (Figure 5A). To understand the role of *SPL12* in osmotic tolerance response, additional experiments were performed on *SPL12*-RNAi and WT alfalfa plants, where phenotypic parameters of plants were recorded. After three weeks of osmotic stress treatment in WT and *SPL12*-RNAi plants, root length, lateral root number, and main root number were affected to various degrees depending on the genotype (Figure 5C–E). *SPL12*-RNAi plants appeared to tolerate stress better than WT because, after three weeks of stress, viable green leaves were observed in *SPL12*-RNAi plants but not in WT (Figure 5B). To investigate possible differences in their ability to grow under osmotic stress, the difference in root length before and after stress was measured. Only WT showed a decrease in root length due to osmotic stress, whereas the *SPL12*-RNAi plants maintained root growth (Figure 5C). Maintenance of root growth by *SPL12*-RNAi also included the number of adventitious (main) roots regenerated from the stems under osmotic stress. In fact, the number of main roots in *SPL12*-RNAi plants did not show any significant changes in comparison between the two conditions (control and osmotic stress), while WT plants showed a reduction over the three weeks of stress compared to the well-watered WT plants (Figure 5D). Furthermore, relative to WT, an increase in lateral root numbers was observed in one of the *SPL12*-RNAi genotypes (RNAi12-7) under the control condition, and in all of the *SPL12*-RNAi transgenic plants under stress (Figure 5E).

### 2.5. SPL12 Silencing Mitigates Nodulation Inhibition under Osmotic Stress

To gain an insight into the function of SPL12 in nodulation under osmotic stress, two weeks after cutting, the rooted *SPL12*-RNAi transgenic plants were inoculated with *S. meliloti* and also treated with mannitol (400 mM) for three weeks (21 dai). When comparing the number of nodules in well-watered (treated with distilled water) and mannitol-treated plants (Figure 6A), WT plants showed a decrease in nodule numbers while *SPL12*-RNAi genotypes maintained nodulation after three weeks of osmotic stress (Figure 6B).

Considering the improved nodule numbers in *SPL12*-RNAi plants at 14 dai [29], we also tested the nodulation capacity of *SPL12*-RNAi plants at 14 dai under osmotic stress. In line with this, under the well-watered condition, *SPL12*-RNAi transgenic plants produced significantly more nodules compared to WT. Following 400 mM mannitol treatment, the nodule number was reduced in WT compared to well-watered condition (Figure 6C), but transgenic *SPL12*-RNAi plants maintained nodulation, in fact, nodule numbers was not noticeably affected by 400 mM mannitol treatment in these transgenic plants (Figure 6C).

### 2.6. Effect of Osmotic Stress on Expression of SPL12, AGL21, and AGL6 in SPL12-RNAi Alfalfa 

To shed light on molecular events associated with SPL12 function under stress conditions, we investigated the effect of mannitol treatment on the transcript levels of *AGL6, AGL21* (regulated by SPL12) and *CLE13* (that inhibits nodulation [47]) in WT and *SPL12*-RNAi alfalfa. The results showed that there were significant differences between plants under stress and control conditions (Figure 7). As expected, the transcript level of *AGL21* was significantly higher in all of the *SPL12*-RNAi plants compared to WT under the control condition (Figure 7A). *AGL21* was also significantly upregulated under stress in two *SPL12*-RNAi genotypes compared to WT. However, *AGL21* was downregulated in WT plants under stress, whereas two of the *SPL12*-RNAi genotypes showed no significant differences (RNAi12-24 and RNAi12-29) compared to plants grown under control conditions (Figure 7A). For *AGL6*, significantly lower transcript levels were detected in WT and *SPL12*-RNAi transgenic plants under the stress condition compared to counterpart plants grown under the control condition (Figure 7B). However, no significant changes in *AGL6* transcript levels were detected in *SPL12*-RNAi genotypes (except for RNAi-24) compared to WT under either control or osmotic stress conditions (Figure 7B).

Given that, under osmotic stress, *SPL12*-RNAi plants at 21 dai produced more nodules compared to WT (Figure 6B), we analyzed the transcript levels of *CLE13* and found a decrease in transcript levels under osmotic stress in all genotypes (Figure 7C). Under the control condition, *CLE13* transcript levels were higher in *SPL12*-RNAi plants relative to WT, while under osmotic stress, there was no significant change in transcript levels of *CLE13* (Figure 7C). This result is consistent with results on nodulation in *SPL12*-RNAi and WT at 21 dai, where there was no significant difference between WT and *SPL12*-RNAi plants, *CLE13* was significantly upregulated in the *SPL12*-RNAi plants at 21 dai [29].

### 2.7. Sulfate Transporters Are Enhanced in SPL12-Silenced Plants

There is a high demand for sulfur in nodulating legumes, and in fact, nitrogen fixation is more sensitive to sulfur deficiency than to nitrate uptake [12,48]. A good supply of sulfur enhances nodulation and nitrogen fixation [12,13]. RNA-seq analysis from a previous study [29] revealed that two Group3 *SULTR* genes, *SULTR3.4* and *SULTR3.5*, were significantly upregulated in *SPL12*-RNAi plants, a finding that was validated by RT-qPCR (Figure 8A,B). 

Since *SULTR3.4* and *SULTR3.5* are members of Group3 *SULTRs,* which are strongly regulated by abiotic stress in plant roots [17], we aimed to investigate their transcript levels under osmotic stress. WT alfalfa plants had higher *SULTR3.4* levels under osmotic stress compared to the unstressed control, and there was a change in transcript levels between treatments in *SPL12*-RNAi plants (Figure 8C). It was noted that *SULTR3.4* expression in RNAi12-7 and RNAi12-29 was higher than in WT under control conditions. WT and RNAi12-29 plants showed a decrease in *SULTR3.5* abundance in response to osmotic stress, whereas RNAi12-7 and RNAi12-24 plants were able to maintain their levels of *SULTR3.5* (Figure 8D). When considering the plants under the stress condition only, RNAi12-7 and RNAi12-24 had an enhanced *SULTR3.5* transcript level compared to WT. *SULTR3.5* expression in well-watered *SPL12*-RNAi plants was higher than in WT, and also relative to counterparts under osmotic stress (Figure 8D). 

### 2.8. Effect of Mannitol Treatment on Expression of Stress-Related Genes

The effect of drought on the expression of antioxidant-related glutathione synthase (*GSH*) [49] and the stress-responsive transcription factor *WD40–1* [50] was previously reported in alfalfa. Enhanced levels of *GSH* and *WD40-1* in miR156-OE alfalfa under drought stress in leaves and roots, respectively, were also previously reported [20,21]. In the current study, the transcript abundance of *GSH* and *WD40–1* was examined to determine whether SPL12 serves to maintain the transcript levels of these genes in alfalfa exposed to osmotic stress. While the transcript levels of *GSH* were higher in well-watered RNAi12-7 and RNAi12-24 compared to WT plants (Figure 9A), they did not show a change in response to osmotic stress in *SPL12*-RNAi and WT plants. In fact, *GSH* was at lower levels in RNAi12-7 under stress relative to control (Figure 9A).

Similarly, for *WD40-1* transcript levels, SPL12i-7 and SPL12i-29 showed higher levels under control treatment compared to WT plants (Figure 9B), but there was no change between *SPL12*-RNAi and WT plants under osmotic stress, with *WD40-1* even showing a decrease in RNAi12-7 and SPL12i-29 under stress relative to the control (Figure 9B). 

### 2.9. AGL6 Silencing Maintains Nodulation under Osmotic Stress

Given that *AGL6* is a direct target of SPL12, and with the observed reduction in transcript levels of *AGL6* in *SPL12*-RNAi genotypes during osmotic stress, we set out to investigate the potential role of AGL6 in nodulation under this stress. Two-week-old rooted WT and *AGL6*-RNAi transgenic plants (L9, L13A, and L13B) were inoculated with *S. meliloti* and treated with mannitol (400 mM) for two weeks (14 dai) or three weeks (21 dai). The number of nodules was compared in both the control and mannitol-treated plants (Figure 10A). 

At 14 dai, *AGL6*-RNAi transgenic plants produced significantly more nodules compared to WT under the well-watered condition (Figure 10B). Upon treatment with 400 mM mannitol, the nodule number was reduced in WT, but *AGL6*-RNAi plants maintained nodulation after two weeks of osmotic stress (Figure 10B). At 21 dai, stressed WT plants had a reduced nodule number when compared to the well-watered WT and the stressed *AGL6*-RNAi plants, while *AGL6*-RNAi genotypes maintained nodulation after three weeks of stress (Figure 10C), thus confirming the likely involvement of *AGL6* in regulating nodulation under osmotic stress in alfalfa.

## 3. Discussion 

### 3.1. How Nitrate Availability Affects Nodulation through the SPL12-AGL21-Regulatory Pathway

To conserve energy, plants inhibit nodulation under conditions of nitrate abundance in the rhizosphere, resulting in a decrease in nodule numbers, nodule mass, and nitrogen fixation, as well as an acceleration of nodule senescence [44]. This regulation of nodulation by nitrate is a part of the AON signaling pathway [43,51]. As the *SPL12*-RNAi and *AGL6*-RNAi plants showed an increase in nodulation, we tested the relationship between nitrate and the miR156/SPL12 regulatory system. Under nitrate-sufficient conditions, the rhizobia-inoculated roots of the *SPL12*-RNAi plants developed more active nodules relative to WT, demonstrating the role of the miR156/SPL12-mediated system in controlling the rhizobia–alfalfa symbiosis. In *Phaseolus vulgaris* (the common bean), miR172c acts as a signaling component of the nitrate-dependent AON, and decreases the sensitivity of nodulation to inhibition by nitrate. Common bean plants overexpressing miR172 showed an increase in active nodules in the presence of nitrate [52]. *At*SPL9 was shown to be a potential nitrate-regulatory hub in Arabidopsis, where it may target primary nitrate-responsive genes [53]. *At*SPL9 expression is affected by nitrate, and the transcript levels of *AtNRT1*.*1*, *AtNIA2*, and *AtNiR* significantly increased in response to nitrate in *AtSPL9* overexpression Arabidopsis plants [53]. In tomato plants (*Solanum lycopersicum*), it was reported that an SPL transcription factor, LeSPL-CNR, directly binds to the promoter of *SlNIA*, resulting in its repressing its expression and activity [54]. LeSPL-CNR was further shown to negatively regulate *SlNIA* transcription levels in response to cadmium (cd) stress in tomato plants [54]. 

Based on our findings, we propose that SPL12 regulates nodulation under a high concentration of nitrate in alfalfa by downregulating *AGL21*. Here, we showed that *AGL21* is upregulated in *SPL12*-RNAi alfalfa plants. *AGL21* is an ANR1 MADS box protein-coding gene, and *At*ANR1 MADS box proteins were previously shown to mediate the effect of externally applied nitrate on lateral root development in Arabidopsis [39,40]. In rice, two MADS box genes, *OsMADS25* and *OsMADS27*, are involved in the regulation of root development in response to nitrate [55]. In Arabidopsis, *AtAGL21* is expressed in different tissues, but most strongly in roots, where *At*AGL21 plays an important role in lateral root development under nitrogen deficiency [41]. In the common bean, *PvAGL21* is expressed in nodules, and its expression is higher in roots compared to pods, seeds, and stems [56]. These results are consistent with our finding that alfalfa *AGL21* is highly expressed in the roots and that its expression is induced by nitrate. Future research should focus on generating and analyzing *AGL21-*silencing and overexpressing alfalfa plants to determine the effect on root architecture, nodulation and nitrogen fixation.

### 3.2. Role of SPL12 and AGL6 in Regulating Nodulation under Osmotic Stress in Alfalfa

Legume crops can adjust their root architecture in response to environmental conditions, not only by branching out, but also by forming a symbiosis with rhizobial bacteria to form nitrogen-fixing nodules [57]. Soil salinity is a major abiotic stress that causes nutrients to become unavailable to plants, and it leads to a nutrient-deprived situation, or nutrient stress, affecting plant yield and root growth (reviewed by Sindhu et al. [58]). Not only does miR156 regulate nodulation in alfalfa, it is also involved in the response to abiotic stress. The miR156-mediated regulation of response to drought, heat, and salinity was previously demonstrated in alfalfa [18,19,20,21], where miR156 targets a number of *SPL* genes for silencing by transcript cleavage in [22,23]. Specifically, SPL13, SPL9, and SPL8 have been investigated for their role in drought tolerance in this plant [20,21,25,26]. Downregulating *SPL13*, *SPL9* and *SPL8* in transgenic plants resulted in alfalfa plants that were less susceptible to drought [20,21,25,26]. *SPL12* was shown to be upregulated in response to mild and severe salinity stress in alfalfa, but was suppressed in all miR156-OE genotypes [19]. In this study, we observed a significant increase in the transcript levels of *SPL12* in WT under osmotic stress as opposed to control conditions. The upregulation of *SPL12* under osmotic stress is consistent with a previous report that showed an increase in *SPL13* transcript levels in WT alfalfa plants under drought [20]. 

The roots are the first plant organ to encounter changes in response to a soil water deficit. Studies in Arabidopsis showed initiation and elongation of lateral roots in drought-tolerant genotypes that led to improved water uptake and drought adaptation [59,60]. In this study, a significant increase in root length accompanied by higher lateral root numbers was observed in alfalfa *SPL12*-RNAi plants under osmotic stress (Figure 5A,E). A previous study by Arshad et al. [20] showed increased root length in miR156-OE and *SPL13-*RNAi alfalfa genotypes under drought stress. Moreover, the miR156-SPL10 module was reported to be involved in root development by silencing *AtAGL79* to control root length and lateral root numbers in Arabidopsis [46]. Therefore, it appears that improved root architecture may help *SPL12*-RNAi alfalfa plants to better access water from deeper in the soil under water scarcity conditions. 

The symbiotic interaction between legume plants and rhizobacteria can be negatively impacted by drought, resulting in reduced nodule numbers and diminished nitrogenase activity [61,62,63]. Nitrogenase activity in root nodules of *M. truncatula* was decreased by 18% and 66% after two and four days of water withdrawal, respectively [64]. It has been shown that in *M. truncatula*, both symbiotic plant components and *S. meliloti* bacteria residing in the root nodules adjust their gene expression profiles in response to drought stress [64]. Our results showed a decrease in the nodule numbers in WT plants under osmotic stress conditions, while *SPL12*-RNAi genotypes maintained nodulation under this stress. The transcript levels of *CLE13* decreased under osmotic stress in all genotypes, while they increased in *SPL12*-RNAi plants under control conditions. This is consistent with increasing nodulation under osmotic stress in *SPL12*-RNAi genotypes. The *AGL6* transcript level did not show any significant changes under the control condition, while its expression was increased only in RNAi12-24 under osmotic stress compared with WT. A previous study showed that SPL12 positively regulates *AGL6*. In fact, the transcript level of *AGL6* was increased in *35S::SPL12* genotypes, but no change was detected in *SPL12*-RNAi plants [29]. Given the functional redundancy of some SPLs, silencing only one *SPL* gene may not be sufficient to affect the expression of *AGL6* in *SPL12*-RNAi genotypes. Here, it was shown that *AGL6* transcript levels were also lower under osmotic stress compared to no treatment, resulting in *AGL6*-RNAi genotypes maintaining their nodulation activity. These observations that *SPL12*-RNAi and *AGL*6-RNAi plants maintained nodulation under osmotic stress suggest a role for SPL12/AGL6 in regulating nodulation in alfalfa under osmotic stress. 

In nodulating legumes, sulfur supply plays an important role in symbiotic nitrogen fixation, as sulfur deficiency causes a decrease in nodulation, inhibition of nitrogen fixation, and a slowing down of nodule metabolism [14]. Accordingly, sulfate transport and metabolism also positively affect nitrogen fixation and nodulation [14]. A sulfate transporter in the symbiosomal membrane of *L. japonicus*, *LjSST1*, was the first indication of sulfate exchange between the two symbiotic partners [15]. *LjSST1* is specifically and highly expressed in the nodules, suggesting a crucial role for this protein in the transport of sulfate from the plant to the bacteroids [15]. The *sst1* mutants developed smaller nodules and displayed symptoms of nitrogen deficiency only under symbiotic conditions. The nodules of the *sst1* mutant plants showed a reduction of approximately 90% in the rate of nitrogen fixation [15]. In the current study, two of the Group3 *SULTR* genes, *SULTR3.4* and *SULTR3.5*, were significantly upregulated in *SPL12*-RNAi plant roots compared to WT. *MtSULTR3.5* in *M. truncatula*, a homolog of *LjSST1*, is strongly expressed in nodules [16]. Another study showed that *MtSULTR3.5* expression is strongly upregulated in *M. truncatula* roots subjected to salt stress [17,65]. Of the sulfate transporters, Group3 *SULTRs* specifically operate under abiotic stress conditions, and they are among salt- and drought-responsive genes in both Arabidopsis and *M. truncatula* [17,65,66,67]. Interestingly, *SULTR3.1* and *SULTR3.4* genes are upregulated in the roots of both Arabidopsis and *M. truncatula* plants subjected to drought stress [17]. Given the above findings, we measured the transcript levels of *SULTR3.4* and *SULTR3.5* in alfalfa root tissues under osmotic stress. The maintenance of *SULTR3.4* and *SULTR3.5* transcript levels under osmotic and control conditions in *SPL12*-RNAi roots indicates that SPL12 must be involved in *SULTR3.4* and *SULTR3.5* regulation. Although the five *At*SULTR3 transporters have been functionally characterized in Arabidopsis [68], further studies are still needed to understand the contribution of nodule sulfate transporters to salt stress response in legumes.

## 4. Materials and Methods 

### 4.1. Plant Material and Growth Conditions

*Medicago sativa* L. (alfalfa) clone N4.4.2 [69] was obtained from Daniel Brown (Agriculture and Agri-Food Canada, London, ON, Canada) and was used as the wild-type (WT) genotype. Alfalfa genotypes with reduced expression levels of *SPL12* and *AGL6, SPL12*-RNAi (RNAi12-7, RNAi12-24, and RNAi12-29) and *AGL6*-RNAi (L9, L13A, and L13B), respectively, and *35S::SPL12-GFP* [29] were used in this study. The transgenic alfalfa plants were generated previously in Dr. Hannoufa’s laboratory using the WT clone N4.4.2 [29]. WT and transgenic alfalfa plants were grown under greenhouse conditions at 21–23 °C, 16 h light/8 h dark per day, light intensity of 380–450 W/m^2^ (approximately 500 W/m^2^ at high noon time), and a relative humidity of 56% for the duration of all experiments. Because of the obligate outcrossing nature of alfalfa, WT and transgenic alfalfa were propagated by rooted stem cuttings to maintain the genotype throughout the study. Stem-cutting propagation and morphological characterization of alfalfa plants were carried out as described previously [28]. 

### 4.2. Phenotypic Analysis of Nodule Development

To determine the number of nodules, plants were examined at 14 and 21 days after inoculation (dai) with *Sinorhizobium meliloti* Sm1021. To eliminate potential microbial contamination, equipment was surface-sterilized using 1% sodium hypochlorite, while vermiculite and water were sterilized by autoclaving for 1 h. *S. meliloti* Sm1021 strain was cultured on a yeast-extract broth agar [70] for two days at 28 °C. A single colony was then inoculated in liquid TY medium and incubated at 28 °C to an optical density OD_600_ nm of 1.5. The alfalfa rooted stems were inoculated by applying 5 mL of bacterial suspension or sterilized water (non-inoculated control) as described previously [28] and allowed to grow for two or three additional weeks. 

### 4.3. Nitrate Treatment

To explore if SPL12-related regulation of nodulation is affected by nitrate, the nodulation test was performed upon treatment with this nutrient. WT and *SPL12*-RNAi alfalfa stem cuttings were grown on vermiculite for 14 days and were then inoculated with *S. meliloti* Sm1021 and treated with KCl or KNO_3_. For this, the 14-day-old inoculated transgenic and WT plants were watered with 3, 8, or 20 mM KNO_3_ or KCl twice a week for two or three weeks. The entire experiment was repeated twice under the same growth and nitrate treatment conditions to test the reproducibility of the results. Effects on nodulation were studied by counting the number of the active (pink) nodules.

To investigate whether treatment with KNO_3_ affects expression of *SPL12* and *AGL21* genes, WT and *SPL12*-RNAi alfalfa plants were grown on vermiculite for 14 days, then the plants were transferred to Murashige & Skoog Modified Basal Salt Mixture without nitrogen (M531, PhytoTechnology Laboratories®, KS, USA) liquid media and left overnight at room temperature. For the nitrate signaling test, the samples were treated with 20 mM KNO_3_ for 0, 5, and 24 h, then roots were collected and flash frozen in liquid nitrogen and stored at −80 °C for later transcript analysis of *SPL12* and *AGL21*.

### 4.4. Mannitol Treatment

To investigate whether SPL12 affects nodulation when plants are grown under osmotic stress, WT, *SPL12*-RNAi, and *AGL6*-RNAi alfalfa plants were grown on vermiculite for 14 days and were then inoculated with *S. meliloti* Sm1021 for two days, followed by treatment with mannitol (to mimic osmotic stress). For mannitol treatment, 16-day-old inoculated WT and transgenic plants were watered with 400 mM mannitol or distilled water once a week for two or three weeks. The below-ground phenotypic parameters were measured according to Aung et al. [27]. The phenotypes included in the characterization were number of main roots, lateral roots, and root length. The roots directly emerging from the stem were considered as main roots, while those that emerged from the main roots were counted as lateral roots. Root length was considered as the length of the longest root. The entire experiment was repeated twice under the same growth and osmotic stress conditions to test the reproducibility of the results. Root samples were harvested from *SPL12*-RNAi and WT plants under osmotic and control conditions and were flash frozen in liquid nitrogen and kept at −80 °C for later transcript analysis of *SPL12*, *AGL21*, *AGL6*, *CLE13*, *SULTR3.4*, *SULTR3.5*, *GSH*, and *WD40-1*.

### 4.5. RNA Extraction, Reverse Transcription and RT-qPCR

Different alfalfa tissues (stems, leaves and roots) were collected and flash frozen in liquid nitrogen and stored at −80 °C until further use. Approximately 100 mg fresh weight was used for total RNA extraction using RNeasy Plant Mini-prep Kit (Qiagen, Hilden, Germany, Cat # 1708891) for leaf and stem samples, and Total RNA Purification Kit (Norgen Biotek Corp., Thorold, ON, Canada, Cat # 25800) for roots. Tissue was homogenized using a PowerLyzer^®^ 24-bench top bead-based homogenizer (Cat # 13155) according to the manufacturer’s manual. Approximately 500 ng of Turbo DNase (Invitrogen, CA, US, Cat # AM1907)-treated RNA was used to generate cDNA using the iScript cDNA synthesis kit (Bio-Rad, Hercules, CA, USA, Cat # 1708891). Transcript levels were analyzed by RT-qPCR using a CFX96 TouchTM Real-Time PCR Detection System (Bio-Rad) and SsoFast™ EvaGreen^®^ Supermixes (Bio-Rad Cat # 1725204) using gene specific primers. Each reaction consisted of 2 μL of cDNA template, 0.5 μL forward and reverse gene-specific primers (10 μM each) (Appendix A), 5 μL SsoFast Eva green Supermix and topped up to 10 μL with ddH_2_O. For each sample three or four biological replicates were analyzed, and each biological replicate was tested using three technical replicates. Transcript levels were analyzed relative to three reference genes: *CYCLOPHILIN* (Cyclo) [71], *β-actin* (*ACTB*) [72], and *ACTIN DEPOLYMERIZING FACTOR* (*ADF*) [71,72] (primers are listed in Appendix A).

### 4.6. ChIP-qPCR Analysis

Shoot tips of alfalfa plants overexpressing *SPL12* tagged with *GFP* driven by the *35S* promoter (*p35S:SPL12m-GFP*) were used as materials for ChIP-qPCR analyses, which were performed based on a previously described protocol [29] with the Chromatin Immunoprecipitation Assay kit (Lot:2382621, Millipore, Billerica, MS, USA). Briefly, nuclei were purified from shoot tips that proteins bound to DNA were cross-linked using 1% formaldehyde under a vacuum for 20 min and the mixtures were ground in liquid nitrogen. The chromatin solution was then sonicated twice at power 3 for 15 s on ice into 500–1000 bp fragments using a Sonic Dismembrator (Fisher Scientific, PA, USA). Ab290 GFP antibody was added to the chromatin solution and protein A-agarose beads were added to recover immune complexes. The precipitated DNA was extracted using phenol: chloroform (1:1, *v:v*) and resuspended in distilled water to be used for ChIP-qPCR analysis using qnMsAGL21 as listed in Appendix A. SPL12 occupancy on *AGL21* was estimated by comparing the fold enrichment in *p35S:SPL12m-GFP* and WT plants. A DNA fragment containing a SBP binding consensus was amplified from a *LATERAL ORGAN BOUNDARES-1*, *LOB1*, gene [73] for use as a negative control.

### 4.7. Phylogenetic Tree Construction 

The phylogenetic tree was constructed based on an alignment of the MADS-box domain and using publicly available sequences of *M. sativa*, *M. truncatula,* and Arabidopsis. Amino acids were aligned by visualization and nucleotides were subjected to ClustalW alignment analysis. The phylogenetic tree was constructed using the neighbor-joining method of phylogenetic tree construction using MEGA7 [74].

### 4.8. Statistical Analysis

Statistical analyses were performed using Microsoft Excel spreadsheet software. Pai-wise comparisons were made using a Student’s *t*-test with either equal or unequal variance. The significant differences between sample means for three or more data sets were calculated using the one-way analysis of variance (ANOVA) where appropriate.

## 5. Conclusions 

Our investigation of the SPL12 function revealed that SPL12 and its direct target, AGL6, regulate nodulation under osmotic stress, as plants with reduced *SPL12* and *AGL6* showed an enhanced number of nodules under this stress, resulting in the maintenance of nodulation in *SPL12*-RNAi and *AGL6*-RNAi plants despite the adverse stress conditions. This study, together with the previous observations that miR156-OE plants had increased tolerance to drought [20,21], and improved nodulation and nitrogen fixation [27], provided evidence that miR156-targeted *SPL12* is a regulator of nodulation under osmotic stress in alfalfa. Moreover, maintenance of nodulation by *AGL*6-RNAi suggests a role for AGL6 in the control of nodulation in alfalfa under osmotic stress (Figure 11).

Our results showed that the role of SPL12 in alfalfa is not only restricted to regulating nodulation under normal conditions, but also controls this process under nitrate-sufficient conditions. Rhizobia-inoculated alfalfa roots with reduced levels of *SPL12* were found to develop more active nodules, relative to WT under nitrate-sufficient conditions, demonstrating the role of the miR156/SPL12-mediated system in controlling rhizobia-alfalfa symbiosis. SPL12 regulates nodulation under nitrate treatment in alfalfa by targeting *AGL21*. *AGL21* is an ANR1 MADS box protein-coding gene. *At*ANR1 MADS box proteins were previously shown to mediate the effect of externally applied nitrate on lateral root development in Arabidopsis [39,40]. Previously, RNAseq and gene ontology analysis showed *AGL21* to be upregulated in *SPL12*-RNAi alfalfa roots [29], where its transcript levels were induced by nitrate. As a negative regulator of *AGL21*, *SPL12* silencing upregulates *AGL21* and enhances the production of active nodules under high nitrate condition (Figure 11). We also determined the direct binding of SPL12 to the *AGL21* promoter. Taken together, our results suggest that SPL12 along with *AGL6* and *AGL21* modulate alfalfa nodulation under osmotic stress and sufficient nitrate conditions.

## Figures and Tables

**Figure 1 plants-11-03071-f001:**
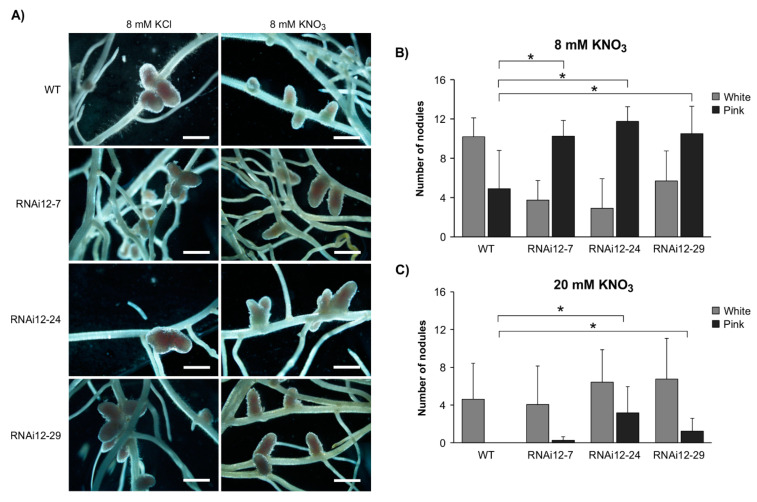
Effect of nitrate on nodule phenotype in *SPL12*-RNAi plants. (**A**) Nodule phenotypes in WT and *SPL12*-RNAi genotypes at 21 dai growing in nitrate-starved media and watered with 8 mM KCl or KNO_3_. Scale bars: 1 mm. The average numbers of pink and white nodules in WT and the *SPL12*-RNAi at 21 dai under (**B**) 8 mM KNO_3_ (n = 15–22 plants) and (**C**) 20 mM KNO_3_ (n = 14–25 plants). * indicates significant differences relative to WT using Student’s *t*-test (*p* < 0.05). Error bar indicates standard deviation.

**Figure 2 plants-11-03071-f002:**
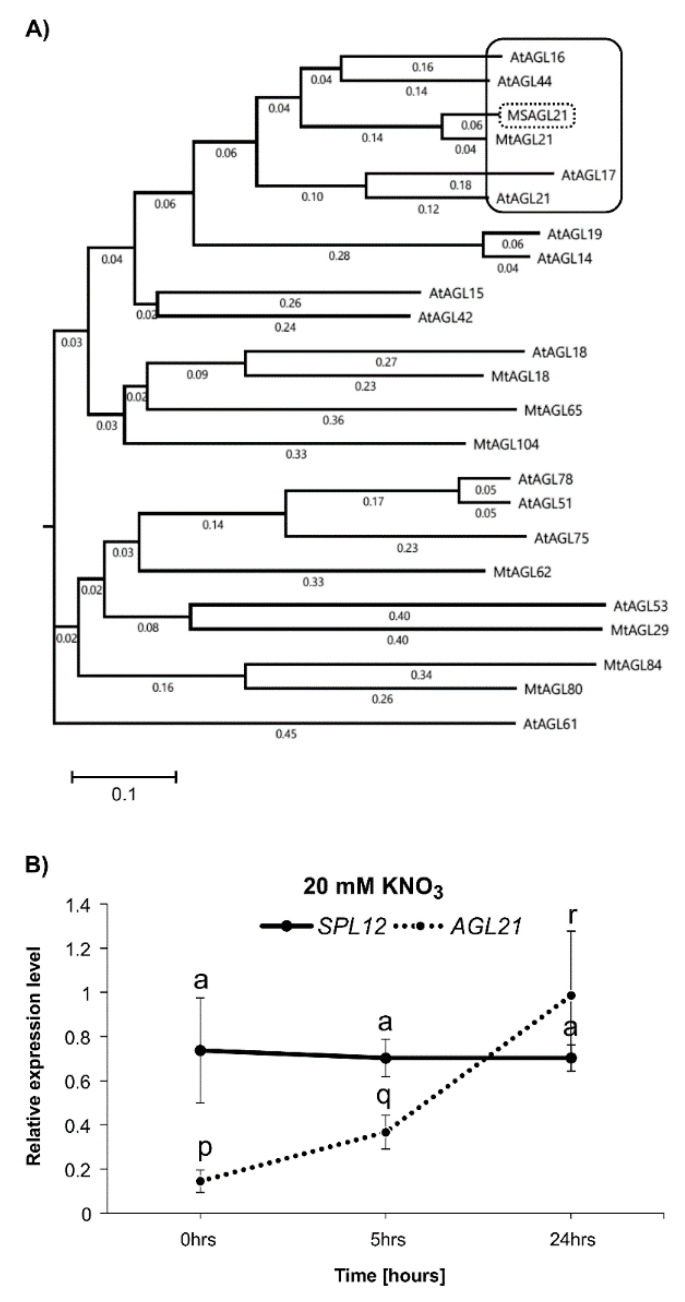
Phylogenetic tree of *M. truncatula* and *Arabidopsis* MADS-box proteins. (**A**) Phylogenetic tree based on protein sequences between *AGL21* and some other MADS-box transcription factors of *M. truncatula* and Arabidopsis. Black frame: ANR1 clade; Black-dashed frame: AGL21 in alfalfa. (**B**) Relative gene transcript levels of *SPL12* and *AGL21* were analyzed in WT at 0, 5, and 24 h after 20 mM nitrate treatment. Statistical groupings across different time points were determined separately for *SPL12* and *AGL21* transcript abundance using one-way ANOVA with the Tukey HSD post hoc test. Significant difference in post hoc Tukey multiple comparisons test is indicated with different letters (“a” for *SPL12* expression level and “p, q, and r” for *AGL21* expression level) Error bar indicates standard deviation.

**Figure 3 plants-11-03071-f003:**
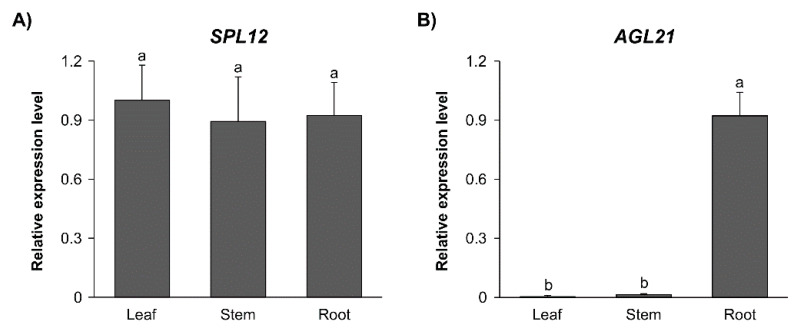
Tissue-specific transcript profiles of *SPL12* and *AGL21*. Relative transcript levels of (**A**) *SPL12*, and (**B**) *AGL2*1 in leaf, stem, and root of WT plants. Significant difference from ANOVA was followed by post hoc Tukey (*p* < 0.05) multiple comparisons test indicated with different letters. Error bars indicate standard deviation.

**Figure 4 plants-11-03071-f004:**
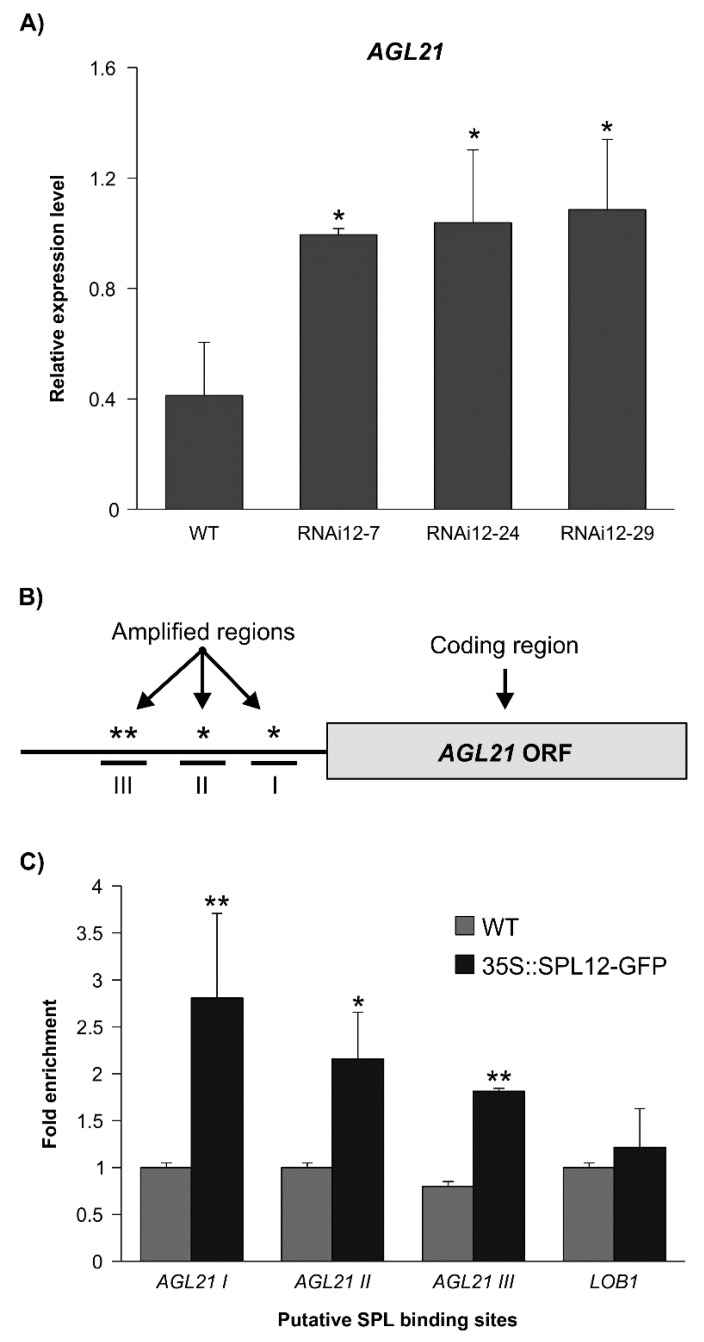
Detection of *SPL12* binding to *AGL21* promoter (**A**) Relative *AGL21* expression in roots of WT and *SPL12*-RNAi alfalfa plants by qPCR. (**B**) Schematic representation of the promoter region of *AGL21*, black box represents the coding sequences (ORF); asterisks indicate locations of putative SPL binding sites on *AGL21* promoter (amplified regions). Roman numerals (I, II and III) indicate the sites that were tested by qPCR. (**C**) ChIP-qPCR-based fold-enrichment analysis of SPL12 in *35S::SPL12m-GFP* and WT plants from means of n = three individual plants where *LATERAL ORGAN BOUNDARES-1, LOB1*, is used as a negative control. * and ** indicate significant differences relative to WT using Student’s *t*-test (n = 3) *p* < 0.05, *p* < 0.01, respectively. Error bar indicates standard deviation.

**Figure 5 plants-11-03071-f005:**
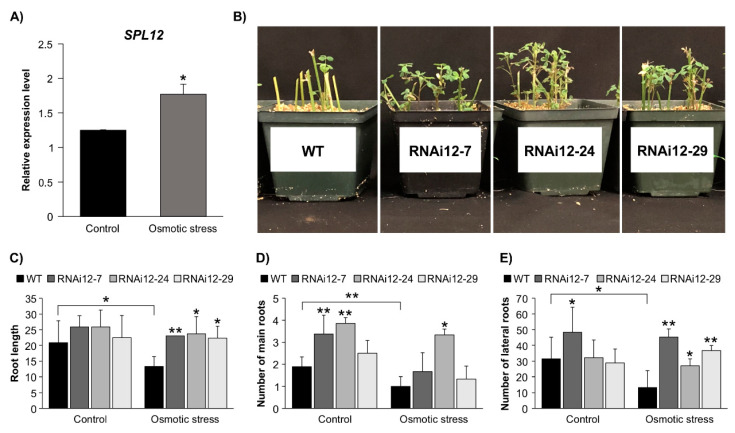
Effect of *SPL12* silencing on response to osmotic stress. (**A**) Relative *SPL12* transcript levels of WT alfalfa treated with mannitol (400 mM) n = 3; (**B**) representative WT and *SPL12*-RNAi plants that were treated with mannitol (400 mM) for three weeks; (**C**) root length; (**D**) number of main roots; and (**E**) number of lateral roots of WT and *SPL12*-RNAi alfalfa under control and osmotic stress conditions (n = 11–14). * and ** indicate significant differences relative to WT using Student’s *t*-test *p* < 0.05, *p* < 0.01, respectively. Error bar indicates standard deviation.

**Figure 6 plants-11-03071-f006:**
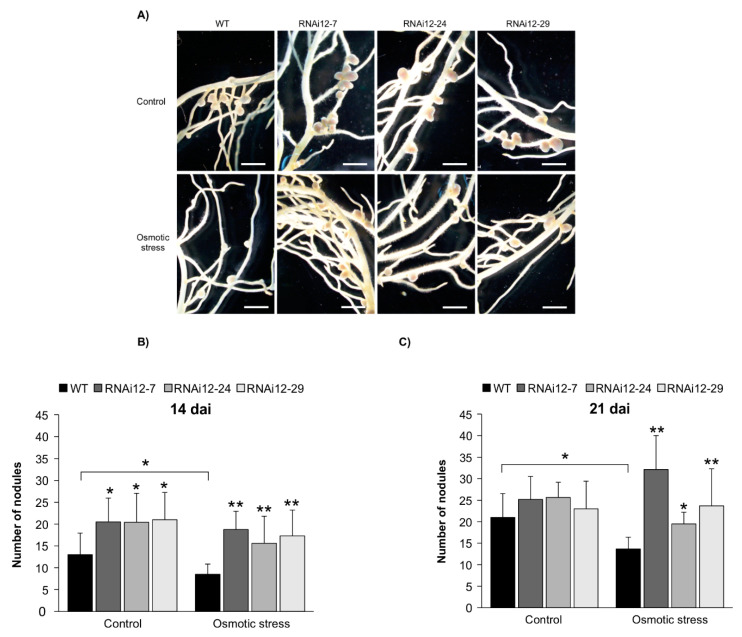
Effect of *SPL12* silencing on nodulation under osmotic stress. (**A**) Phenotypes of nodules of WT and *SPL12*-RNAi plants under osmotic stress (400 mM mannitol) at 21 dai. Scale bars: 2 mm. (**B**) The number of nodules in WT and *SPL12*-RNAi alfalfa plants under control and osmotic stress conditions (n = 12–14) at 21 dai; and (**C**) at 14 dai (n = 10–12 plants). * and ** indicate significant differences relative to WT using Student’s *t*-test *p* < 0.05, *p* < 0.01, respectively. Error bar indicates standard deviation.

**Figure 7 plants-11-03071-f007:**
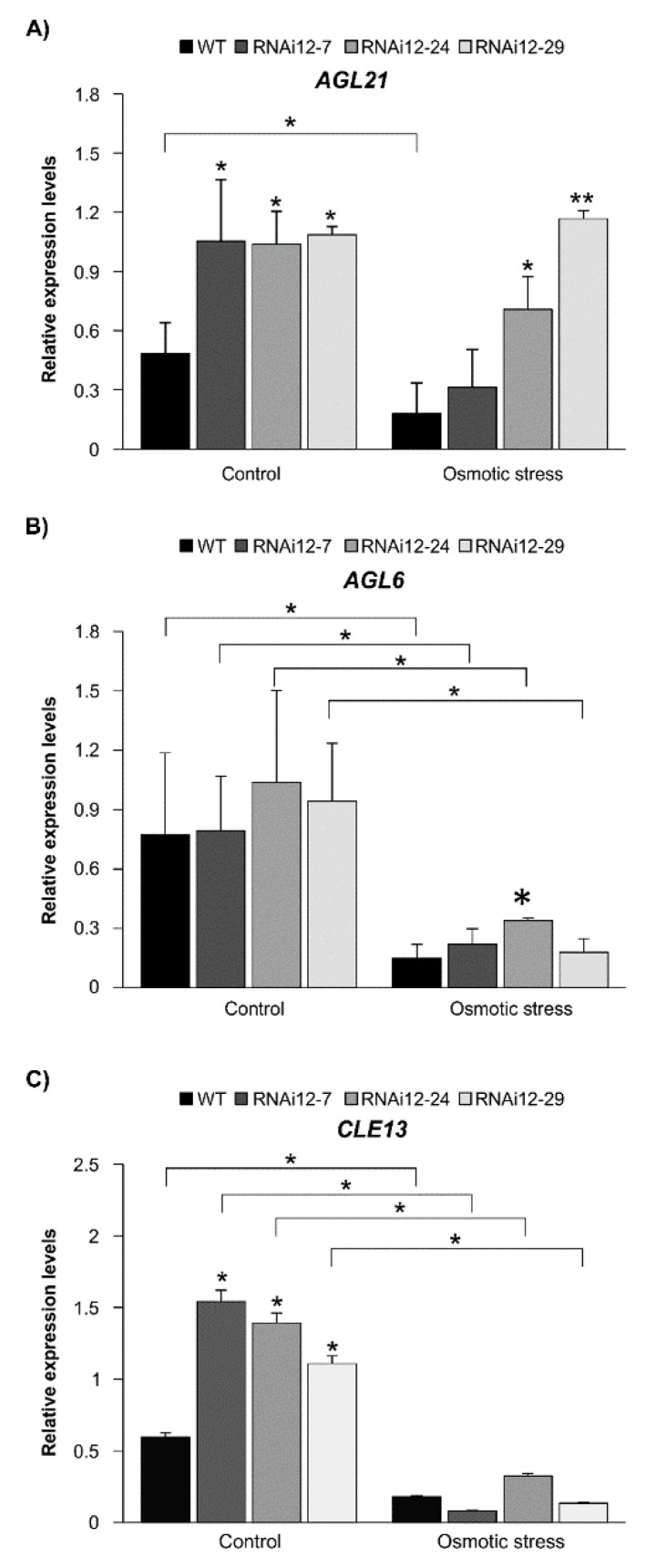
Effect of osmotic stress on transcript levels of *AGL6*, *AGL21,* and *CLE13* in WT and *SPL12*-RNAi alfalfa. (**A**) *AGL21*, (**B**) *AGL6,* and (**C**) *CLE13*. * and ** indicate significant differences relative to WT using Student’s *t*-test (n = 4) *p* < 0.05, *p* < 0.01, respectively. Error bar indicates standard deviation.

**Figure 8 plants-11-03071-f008:**
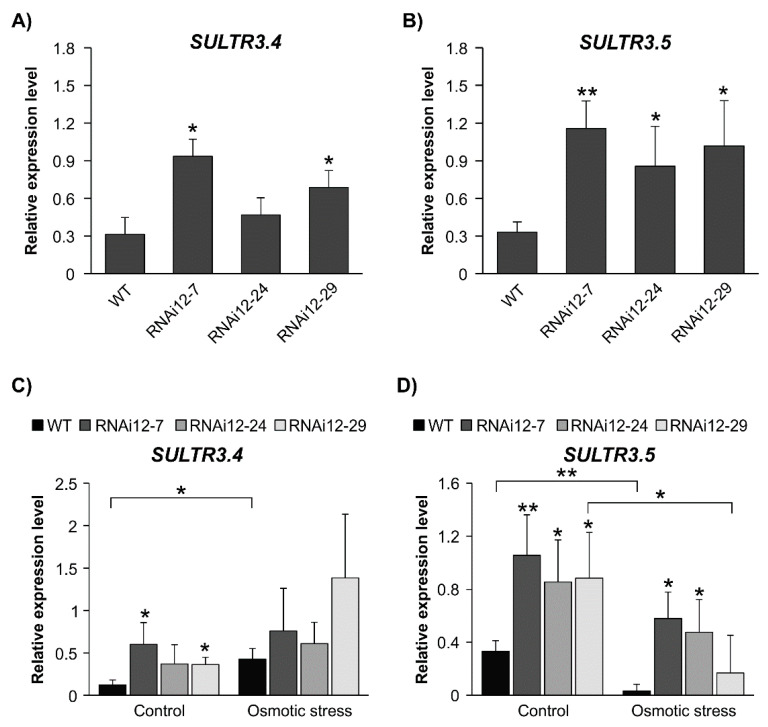
Effect of osmotic stress on transcript levels of sulphate transporter genes in WT and *SPL12*-RNAi alfalfa plants. Relative transcript levels of (**A**) *SULTR3.4* and (**B**) *SULTR3.5* in WT and *SPL12-*RNAi plants. Relative transcript levels of (**C**) *SULTR3.4* and (**D**) *SULTR3.5* in WT and *SPL12*-RNAi alfalfa exposed to three weeks of osmotic stress (400 mM mannitol). * and ** indicate significant differences relative to WT using Student’s *t*-test (n = 3) *p* < 0.05, *p* < 0.01, respectively. Error bar indicates standard deviation.

**Figure 9 plants-11-03071-f009:**
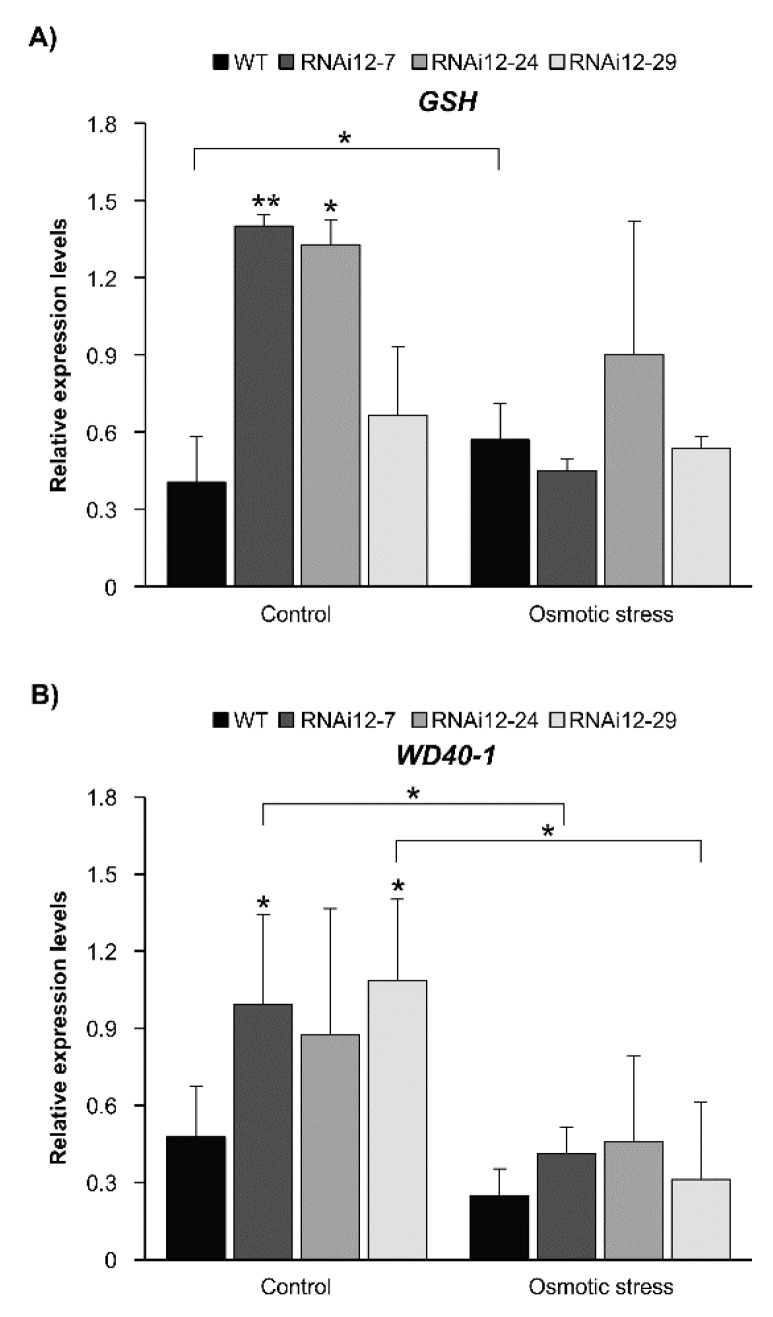
Relative transcript levels of stress-related genes in response to osmotic stress. Transcript levels of (**A**) *GSH* and (**B**) *WD40-1* in WT and *SPL12*-RNAi roots under osmotic (400 mM) and control conditions. * and ** indicate significant differences relative to WT using Student’s *t*-test (n = 3) *p* < 0.05, *p* < 0.01, respectively. Error bar indicates standard deviation.

**Figure 10 plants-11-03071-f010:**
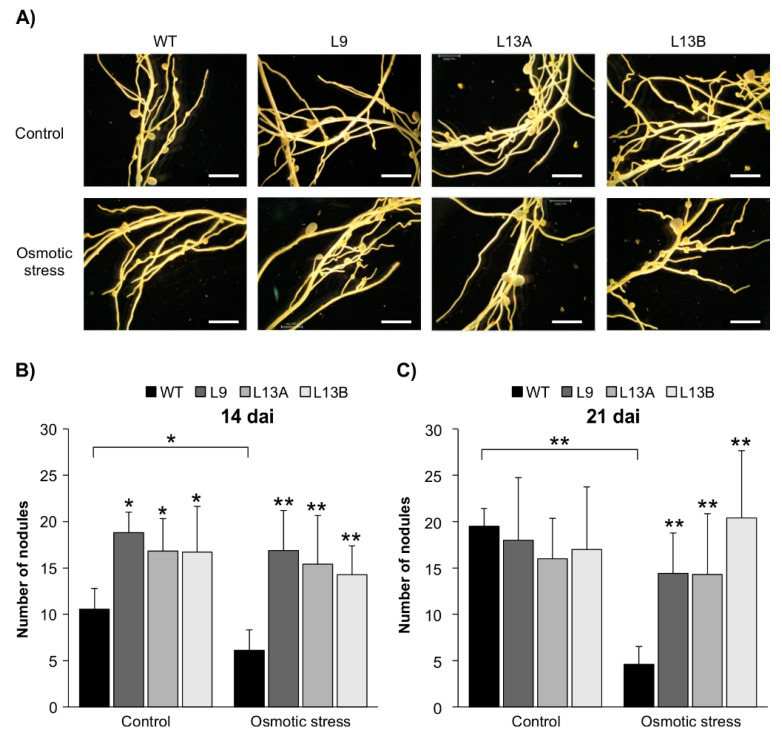
Effect of *AGL6* silencing on nodulation under osmotic stress. (**A**) Nodule phenotypes of WT and *AGL6*-RNAi plants that were exposed to osmotic stress (400 mM) (21 dai). Scale bars: 3 mm. (**B**) The number of nodules in WT and the *AGL6*-RNAi alfalfa under control and osmotic stress conditions (n = 12–15) at 14 dai, and (**C**) 21 dai (n = 8–11 plants). * and ** indicate significant differences relative to WT using Student’s *t*-test *p* < 0.05, *p* < 0.01, respectively. Error bar indicates standard deviation.

**Figure 11 plants-11-03071-f011:**
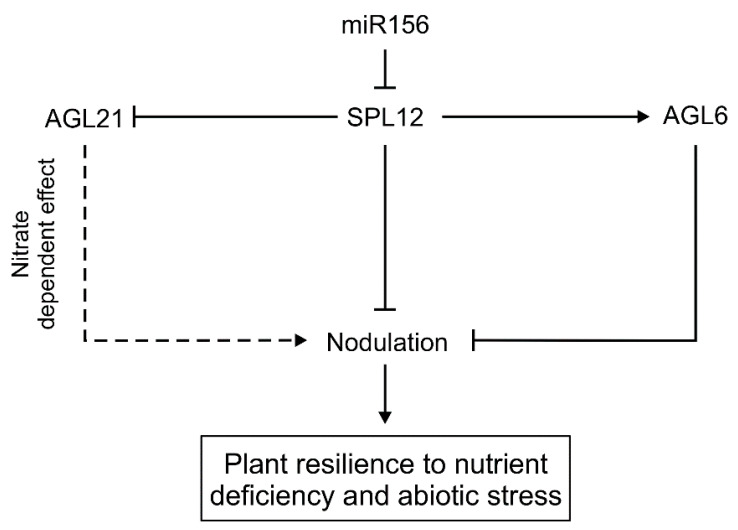
A model showing pathways for miR156/SPL12 regulation of nodulation in alfalfa. Solid lines or arrows indicate an experimentally confirmed mechanism, while dotted lines or arrows show a predicted pathway. Arrow heads indicate positive regulation while line heads indicate negative regulation.

## Data Availability

Not applicable.

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
