# Peer review of "SPL12 Regulates AGL6 and AGL21 to Modulate Nodulation and Root Regeneration under Osmotic Stress and Nitrate Sufficiency Conditions in Medicago sativa"

_plants, 2022, doi:10.3390/plants11223071_

Round 1

Reviewer 1 Report

Comments to the authors:

I’ve reviewed the manuscript titled “SPL12 Regulates AGL6 and AGL21 to Modulate Nodulation and Root Regeneration under Osmotic Stress and Nitrate Sufficiency Conditions in Medicago sativa” which tends to explore the highly conserved plant microRNA, miR156, affects root architecture, nodulation, symbiotic nitrogen fixation, and stress response. The topic is interesting and the results may provide useful information on improving symbiotic nitrogen fixation capacity and increasing biomass with reducing the use of nitrogen fertilizer. However, the study has some flaws and requires major revisions in order to meet the journal’s requirements. My comments are below:

General comments:

1、As for the part of result 2.3, expression profiles of SPL12 and AGL21 in alfalfa, I don't think it has any significance. It is suggested to put it into section 2.2 or remove it directly.

2Figure 5C-E shows that there are significant differences in root development. Therefore, the difference in the number of nodules may be caused by root development. The nodule density may be more appropriate than the number of nodules to express your results. Why not use the density of nodules to explain your results?

3. Figure 6A and Figure 10A show the number of nodules in 21 days, but why put the number of nodules in 14 days? What does Figure 10B mean?

4. Why is the AGL6 expression of RNAi12-24 so high in Figure 7B? As shown in Figure 11, SPL12 can promote AGL6 expression, which is inconsistent with the results in Figure 7B. Why?

My specific comments are below:

1、The 8mM KNO3 below Figure 1a should be placed above Figure 1b. Please modify this picture.

2、Footnotes of all figures should be capitalized ABC, and P-Value should be marked in italics.

3It is inappropriate to write the abscissa of Figure 4c in this way. It is recommended to refer to other documents for modification.

4Figure 5B has no legend, please add.

5The order of Figure 6B and Figure 6C should be changed. It may be more appropriate in chronological order. Besides, why choose these two times? Is there any basis for that?

Reviewer 2 Report

       In the manuscript from Nasrollahi et al., the authors reported that SPL12 can regulate the expression of AGL21 to modulate the nodulation and root growth in Medicago sativa. Their results showed that SPL12 can suppress the expression of AGL21 to prevent the nodulation of alfalfa under high concentration of nitrate. They found the expression of AGL21 can be induced by high nitrate treatment and SPL21 can directly bind to the promoter of AGL21 to regulate its expression. Furthermore, the authors characterized the phenotypic and molecular parameters of SPL12-RNAi and AGL6-RNAi lines under osmotic stress, which revealed that the SPL12/AGL6 module plays a negative role in maintaining nodulation under osmotic stress. Taken together, this research discovered that SPL12 along with AGL21 and AGL6 regulate the nodulation of alfalfa in nitrate-sufficient and osmotic stress condition.

       This manuscript is well-written and easy to follow. However, my immediate feeling of this manuscript is that the idea of this study is not so clear. There is lack of logical connection between each section, even though the authors presented lots of data. Besides, some critical information is missing or lack of resource.

The following are main comments:

1.       There were no line numbers which made it hard to label.

2.       The authors didn’t provide any supporting information that the SPL12 was successfully knockdown in the RNAi12-7, RNAi12-24 and RNAi12-29 lines which is the basic of the research in this manuscript. The real-time PCR results should be provided.

3.        The abbreviation ‘dai’ should be the annotated when it appeared firstly.

4.       The sentence of “Under 20 mM KNO3 treatment, WT plants formed only small white nodules, while RNAi12-24 and RNAi12-29 plants produced significantly more pink nodules (Figure 1c)” is confused. As illustrated in the figure 1C, there are at least 4 white nodules in WT plants, which is inconsistent with “only small white nodules” in the sentence.

5.       The images in figure 1 were not well organized. There was “8 mM KNO3” and half of “ B)” below figure 1A and there were another half of “8 mM KNO3” above figure 1B. please re-arrange the image in this figure.

In the legend of figure 1, there are 2 “(b), but (c) could not be found.

6.       Figure 2B. what do the letters “a”, “p”, “q” and “r” mean in figure 2B?

7.       The sentence of “After three weeks of osmotic treatment, SPL12-RNAi, root length, lateral root number, and main root number were affected by osmotic stress to various degrees depending on the genotype (Figure 5c, d, e)”, should be re-organized. “SPL12-RNAi,”?

8.       The conclusion that “Maintenance of root growth by SPL12-RNAi also included the number of adventitious (main) roots regenerated from the stems under osmotic stress, while WT plants showed a reduction over the three weeks of stress (Figure 5d)” could not be drawn from the figure 5d. According to figure 5d, only RNAi12-24 can maintain main root growth while there was no difference between RNAi12-7, RNAi12-29 and WT in osmotic stress condition. So, it is very important to verify that if SPL12was successfully knockdown in line RNAi12-7 and RNAi12-29, as I talked in comment 2.

9.       In the methods, the author didn’t provide enough information about how SPL12-RNAi and AGL6-RNAi lines to be created and the key sequences for plasmids construction.

10.   Table S1 couldn’t be found in supplementary materials.

Round 2

Reviewer 1 Report

I think it is necessary authors should check carefully one by one to ensure no grammatical errors and inappropriate expression before this manuscript is published.

Reviewer 2 Report

In the revised manuscript from Nasrollahi et al., the authors made the essential modifications, re-organized the figures and text, and provided important data to address all the concerns raised in the first-round reviews. The manuscript has improved a lot that I think it is ready for publication.